# Characterisation of Severe Traumatic Brain Injury Severity from Fresh Cerebral Biopsy of Living Patients: An Immunohistochemical Study

**DOI:** 10.3390/biomedicines10030518

**Published:** 2022-02-22

**Authors:** Ping K. Yip, Shumaila Hasan, Zhuo-Hao Liu, Christopher E. G. Uff

**Affiliations:** 1Centre for Neuroscience, Surgery & Trauma, Blizard Institute, Barts and The London School of Medicine and Dentistry, Queen Mary University of London, London E1 2AT, UK; 2Department of Neurosurgery, Royal London Hospital, Whitechapel, London E1 1FR, UK; shumaila.hasan1@nhs.net; 3Department of Neurosurgery, Chang Gung Memorial Hospital at Linkou, Chang Gung University, Taoyuan 33302, Taiwan; b8402022@gmail.com

**Keywords:** traumatic brain injury, brain biopsy, neuronal injury, dendritic injury, neurovasculature, neuroinflammation, Glasgow Outcome Scale-Extended

## Abstract

Traumatic brain injury (TBI) is an extremely complex disease and current systems classifying TBI as mild, moderate, and severe often fail to capture this complexity. Neuroimaging cannot resolve the cellular and molecular changes due to lack of resolution, and post-mortem tissue examination may not adequately represent acute disease. Therefore, we examined the cellular and molecular sequelae of TBI in fresh brain samples and related these to clinical outcomes. Brain biopsies, obtained shortly after injury from 25 living adult patients suffering severe TBI, underwent immunohistochemical analysis. There were no adverse events. Immunostaining revealed various qualitative cellular and biomolecular changes relating to neuronal injury, dendritic injury, neurovascular injury, and neuroinflammation, which we classified into 4 subgroups for each injury type using the newly devised Yip, Hasan and Uff (YHU) grading system. Based on the Glasgow Outcome Scale-Extended, a total YHU grade of ≤8 or ≥11 had a favourable and unfavourable outcome, respectively. Biomolecular changes observed in fresh brain samples enabled classification of this heterogeneous patient population into various injury severity categories based on the cellular and molecular pathophysiology according to the YHU grading system, which correlated with outcome. This is the first study investigating the acute biomolecular response to TBI.

## 1. Introduction

Traumatic brain injury (TBI) affects millions of patients annually worldwide, and many suffer severe disability and death [1,2]. When a head injury occurs, the brain is subjected to a multitude of forces, many unknown and generally chaotic, resulting in a plethora of injuries (epidural, subdural, subarachnoid and intracerebral bleeding, traumatic diffuse axonal injury, and both coup and contrecoup injury patterns) [3], some of which are predictable and some which are not. Known and unknown factors, including age, handedness, pre-existing medical conditions, anatomical variants, a history of TBI, and genetic predisposition, all influence the course of the disease and therefore outcome [4], resulting in a massive variation in functional outcome stemming from seemingly similar injury mechanisms and clinical presentations.

Intensive care can support the unconscious patient and deep anaesthesia reduces the metabolic requirements of the brain, allowing it to tolerate reduced blood flow and higher intracranial pressure (ICP) [5]. Surgery can reduce ICP by evacuating haematomata, draining cerebrospinal fluid (CSF), or decompressive craniectomy, all with the intent of mitigating secondary brain injury. However, no therapeutic agent has been shown to alter the course of the primary brain injury, despite more than 50 trials of 31 therapies at a cost of $1.1 billion since 1993 [6].

TBI is usually classified as mild, moderate, or severe by various parameters, including clinical severity (including imaging), pathoanatomic type, outcome, and prognosis [7,8]. Although it has been recognised that multiple factors, including age and comorbidities, affect outcome in TBI [4], these are not considered in any current TBI classification systems. It has also been identified that the ageing process is accelerated after TBI [9]. Outcome measures in clinical trials of TBI are generally dichotomised, and these systems often fail to capture sufficient detail of this complex and varied disease, which may explain the massive difference in outcomes observed in seemingly similar injuries [10].

Oncology has, for years, benefited from accurate diagnosis based on histological examination, allowing arduous and unpleasant treatments to be confidently recommended or withheld. However, TBI does not benefit from anything comparable. In the early stages of TBI, prognosis is often uncertain, particularly with respect to whether any meaningful recovery will occur. Even if the likely outcome is thought to be a very poor neurological recovery, care is frequently continued for months in the hope that recovery will be better than expected. Furthermore, treatments such as decompressive craniectomy and tracheostomy may be undertaken without robust evidence.

The Glasgow coma scale (GCS) has been a mainstay of classification and prognostication for more than 40 years [11]; however, factors such as intoxication and sedation are confounders which can result in poor correlation [12]. Furthermore, it is possible that the universal failure of clinical trials of therapeutic agents in TBI—all of which showed promise in preclinical and/or phase I and II clinical trials—was, in part, due to inaccurate classification systems which lacked adequate resolution, resulting in dilution of results [13].

Medical imaging, while currently the cornerstone of TBI diagnosis, is yet unable to detect important microscopic cellular changes due to a lack of spatial resolution. TBI can lead to various types of pathophysiological events, such as neuronal injury, dendritic injury, neurovascular disruption, and neuroinflammation, [14,15,16]. These cellular and molecular changes occur at microscopic levels, and by the time clinical deterioration such as raised ICP, pupillary abnormalities, or neuroimaging changes occur, it is often too late for effective treatment.

A better method of identifying the pathophysiology of TBI at the microscopic level is therefore required. Immunohistochemistry is a popular technique used universally in most scientific fields. During the past decade, the knowledge of neuronal injury, dendritic injury, neurovascular injury, and neuroinflammation has vastly increased based on in vitro, in vivo and post-mortem studies. For example, neuronal injury has been studied using neuronal nuclear protein (NeuN) in TBI [14,16,17], and dendritic injury using microtubule-associated protein-2 (MAP2) in rodent cortical injury [18]. Neurovascular injury in TBI has been studied using claudin-5 and von Willebrand factor (vWF) [19]. Neuroinflammation in traumatic CNS injuries, particularly in microglia, has been studied using ionised calcium-binding adaptor molecule 1 (Iba1) and/or P2Y12 [16,20].

Brain biopsy remains the gold standard of diagnosis in neuro-oncology [21] and has been used in other conditions, including presenile patients [22,23] and normal pressure hydrocephalus [24]. A single study performing brain biopsy in the superior frontal gyrus (the standard position for ICP monitors) prior to ICP monitor insertion in severe TBI demonstrated the safety of the procedure [25]. In addition to this study, which demonstrated a global TBI-associated proteomic response, there has been evidence in animal models that severe TBI results in global biomarker (Galectin-3) expression throughout the brain [16,26]. We therefore aimed to investigate the global cellular and molecular response to severe TBI within the brain using a standardised location; biopsies obtained at craniotomy were also taken from the superior frontal gyrus.

The aims of this study were to affirm the safety of brain biopsy in severe TBI, and to determine the various pathophysiological responses after TBI at the microscopic level using immunohistochemistry (IHC) in fresh brain biopsies. The study, which was locally called Severe Head Injury Brain Analysis (SHIBA), was able to classify certain groups of brain injury pathologies that could diagnose and prognosticate severe TBI patients based on the expression of a variety of immunostaining markers.

## 2. Materials and Methods

Ethical approval was granted by the London-Camden & Kings Cross Research Ethics Committee (REC ref: 20/LO/0074) and this application was supported by a patient and public involvement project where >90% of participants were supportive of brain biopsy in TBI [27]. All patients were unconscious at the time of enrolment and in line with the declaration of Helsinki, the 2005 UK Mental Capacity Act and the REC approved protocol, the next of kin were asked to act as a personal consultee or, if none were available, a senior doctor was asked to act as an Independent Healthcare Professional (IHP) consultee. Twenty-five consecutively eligible severe TBI patients were recruited at the Royal London Hospital, London, UK, between July 2020 to May 2021. Two patients recovered to give their own consent prior to discharge.

### 2.1. Patients

Inclusion criteria comprised age ≥ 18 years and TBI requiring diagnostic or therapeutic breaching of the dura as part of standard care. Exclusion criteria included coagulopathy (point of care or laboratory international normalised ratio (INR) testing was performed in all patients) and current use of anticoagulant or antiplatelet medication. Safety was assessed by follow up imaging performed as part of standard care (i.e., extra scans were not performed). All 25 patients were followed up clinically for 6 months.

### 2.2. Tissue Collection and Processing

Brain tissue was sampled opportunistically if the dura was opened for diagnostic (ICP monitoring or external ventricular drain (EVD) insertion) or therapeutic (craniotomy) purposes. When sampling during ICP monitor insertion, a Kocher’s point twist drill craniostomy was fashioned, generally on the right side, and the dura perforated to allow insertion of the intraparenchymal ICP transducer. Prior to insertion of the transducer, a biopsy cannula (Neuromedex GmbH, Vierenkamp 15, 22453 Hamburg, Germany, ref 79.2502110) was introduced to a depth of 1 cm and a single core biopsy of brain tissue (10 mm × 1.5 mm) was taken prior to the ICP monitor being inserted. This method was also employed for tissue sampling during insertion of an EVD. During craniotomy, a single core biopsy was taken at a depth of 1cm from the superior frontal gyrus using a biopsy needle. The brain biopsy tissue was immediately placed into 10% formalin (Figure 1A). After at least 2 h, the sample was transferred into 20% sucrose in 0.1 M phosphate-buffered solution and stored at 4 °C until further processed.

If acute subdural haematoma was present and if brain tissue or brain contusions were resected for therapeutic purposes, these samples were collected as additional specimens in 10% formalin and processed as above. Additionally, blood, cerebrospinal fluid (CSF) (if being drained therapeutically via an external ventricular drain (EVD)), saliva, urine and faeces were collected for 7 days when available. Analysis of these samples is beyond the remit of this study and will be reported separately.

### 2.3. Immunohistochemistry

A small sample of the brain biopsy, representing the dorsal region of the cerebral cortex, was embedded into optimal cutting temperature (OCT) compound and then quickly frozen using dry ice. Using a freezing cryostat, the tissue was consecutively cut cross-sectionally into 12 µm thick sections and thaw-mounted onto microscope slides (Trajan Series 3, Medline Scientific, Chalgrove, UK). After at least one hour, the slides were further processed for immunohistochemistry as previously described [20]. Briefly, the slides were washed for 3 × 5 min with 10 mM phosphate-buffered saline (PBS), then incubated in antigen unmasking solution according to manufacturer’s protocol at 80 °C for 30 min. After at least 10 min of cooling at room temperature, the slides were washed with PBS for 3 × 5 min before being incubated with 2% skimmed milk (Marvel) for at least 30 min. Thereafter, the 2% skimmed milk was decarded and primary antibodies were applied. The primary antibodies used were: mouse anti-NeuN (1:500, #MAB377, Chemicon, Watford, UK), mouse anti-MAP2 (1:500, #MAB3418, Merck Millipore, Watford, UK), mouse anti-claudin-5 (1:100, #35-2500, ThermoFisher, Horsham, UK), rabbit anti-vWF (1:200, #AB7356, Merck Millipore, Watford, UK), rabbit anti-Iba1 (1:500, # 019-19741, Wako, Alpha Laboratories Ltd, Hampshire, UK), and rabbit anti-P2Y12 (1:400, #ANA55043A, AnaSpec, Cambridge Bioscience, Cambridge, UK). Primary antibodies were allowed to incubate on the tissue sections overnight at room temperature in a sealed staining box. The following day, the slides were washed in PBS 3 × 5 min. Then, secondary antibodies were applied onto the sections and allowed to incubate for two hours at room temperature. The secondary antibodies used were: donkey anti-rabbit Alexa Fluor^®^ 488 or 594 (1:500, #A21206, #A21207, ThermoFisher, Horsham, UK), or donkey anti-mouse Alexa Fluor^®^ 488 (1:500, #Ab150105, Abcam, Cambridge, UK). Thereafter, the secondary antibodies were discarded. Hoechst (1:1000) was added for five minutes to counterstain the nuclei, then washed with PBS for 3 × 5 min before being cover-slipped using Vectashield^®^ mounting medium (H-1000-10, Vector Labs, 2Bscientific Ltd, Upper Heyford, UK). Negative controls were treated the same as the experimental slides but with the absence of the primary antibody (Figure 1D). The slides were visualised with a fluorescent microscope (Figure 1B,C) and images were captured at ×20 and ×40 magnification using a Kern ODC825 microscope digital camera and ToupView v1.0 software (Hangzhou ToupTek Photonics Co., Ltd, Zhejiang, China).

The cellular and molecular expressions were further characterised in samples from 3 patients in each group of GOS-E severity. The 3 groups were: GOS-E 7–8 (good recovery), GOS-E 4 (moderate disability) and GOS-E 1 (died). The primary antibodies used were: rabbit anti-synaptophysin (1:500, #5461, Cell Signalling Technology, London, UK), mouse anti-AT8 (1:500, #MN1020, ThermoFisher, Horsham, UK), rabbit anti-VE Cadherin (1:100, #LS-2138, LifeSpan BioSciences, 2Bscientific Ltd, Upper Heyford, UK), rat anti-CD16/32 (1:500, #553141, BD Pharmingen, Wokingham, UK), goat anti-arginase-1 (1:250, # SC-18354, Santa Cruz Biotechnology, Inc, Insight Biotechnology Ltd, Wembley, UK), and rabbit anti-GFAP (1:1000, #Z0334, Dako, Stockport, UK).

Analysis of the further immunostaining was carried out using ImageJ software with a customised macro, similar to previously published methods [16,20]. Briefly, two regions of interest (ROI) per section were randomly selected and the intensity was measured. In double immunostained sections, the ROI identified for P2Y12 was superimposed onto the same image to detect coexpression. The percentage cell count of reactive microglia was the number of classically activated microglial cells divided by the total number of P2Y12 immunostained microglial cells.

### 2.4. Data Analysis and Statistics

The stained brain sections were independently evaluated blind to the patient’s clinical condition by a neuroscientist (P.K.Y.) and an academic neurosurgeon (Z.H.L.), who have more than thirty combined years of histological experience with immunohistochemistry in traumatic CNS injuries, involving neuronal injury [20,28], dendritic injury [29], neurovascular injury [15], and neuroinflammation [16,20]. The Glasgow Outcome Scale-Extended (GOS-E) at 6 months post injury was used to define the outcome of the patients as it has been validated as a good indicator of severity, disability, cognitive dysfunction, and health in TBI patients [30,31,32].

Linear regression analysis was used to determine the linear relationship between the YHU grade of various pathological changes and the GOS-E score. Normality analysis using the Shapiro–Wilk test was used to determine whether parametric or non-parametric statistical tests should be used. Unpaired t-tests were used to determine significant differences between the total YHU grade, distance from injury to biopsy site and method of biopsy (ICP monitor or craniotomy). A one-way ANOVA followed by post-hoc Tukey test was used to determine significant differences in immunostaining for: synaptophysin, tau, GFAP, VE cadherin, and co-expression of P2Y12 and Arginase-1 or CD16/32 between the 3 GOS-E groups. All statistical analyses and graphical representations were performed using GraphPad Prism v8. Data were expressed as mean ± standard deviation (SD) for clinical data and mean ± standard error means (SEM) for histological data. Statistical significance was determined when the *p* value < 0.05.

## 3. Results

### 3.1. Demographics of Patients

Of the 25 TBI patients recruited in the SHIBA study, 22 patients were male and 3 were female (Table 1). The mean (±SD) age of patients was 41.2 years (±16.1) and the mean Glasgow Coma Scale (GCS) was 6.9 (±3.6) at presentation (Table 1). Injury mechanisms included fall (44%), road traffic accident (32%), assault (12%), gunshot wound (8%), and train accident (4%).

As previously stated, all patients were unconscious due to a head injury and had been intubated. Their injuries were subsequently classified according to the Abbreviated Injury Scale (AIS); 21 patients had an AIS of 5, 2 had an AIS of 4, one had an AIS of 3, and one had an AIS of 2. Patients with AIS 2 and 3 were intubated pre-hospital due to history of head injury, decreased GCS, and agitation.

At 6 months post injury, the mean (±SD) Glasgow Outcome Scale-Extended (GOS-E) was 4.0 (±2.7), dichotomised between GOS-E scores of 4 and 5, including 64% unfavourable versus 36% favourable outcomes, respectively (Table 1). Of the unfavourable outcomes, GOS-E 1 (death) occurred in 28%, GOS-E 2 (vegetative state) in 12%, GOS-E 3 (lower severe disability) in 12%, and GOS-E 4 (upper severe disability) in 12%. In contrast, of the favourable outcomes, GOS-E 5-6 (moderate disability) was seen in 4%, and GOS-E 7–8 (lower or upper good recovery) in 32%. Of the 7 patients who died, the mean (±SD) survival was 9.9 (±10.7) days, ranging from 2 days to 33 days post injury (Table 1).

The patients had very few comorbidities and the protocol appears to have introduced an unintentional bias towards this by excluding those taking anticoagulant or antiplatelet medication. All the patients who did not survive died from their head injury (as opposed to death from other injuries). A single patient also had COVID pneumonia (as will be discussed below) but apart from this, there were no patients where comorbidity (as opposed to TBI) was considered to have contributed to the outcome.

### 3.2. Biopsy Sampling

Fresh brain biopsies were obtained from the left (28%) or right (72%) superior frontal gyrus. Biopsy was performed prior to ICP monitor insertion in 48%, prior to EVD insertion in 8%, and at craniotomy in 44%. In 11 patients (44%), the biopsy was taken from a site remote to the primary (coup) or contrecoup injury. Mean (±SD) sampling time was at 14 h (±31), with a mode of 1 h: 32% were carried out within 1 h, 20% within 8 h, 8% within 24 h, and 12% greater than 48 h after injury. We aimed to sample as soon as possible after injury and this range includes patients who suffered clinical deterioration while in hospital or who had a long lie prior to being found.

Twenty-four patients had an additional CT scan within 24 h of the biopsy as part of standard medical care. One patient recovered rapidly and there was no clinical indication for further imaging, although a delayed MRI scan showed no evidence of haemorrhage at the biopsy site. Punctate haemorrhage was observed at the biopsy site in two patients (8%). In one patient, a significant blossoming of frontal contusions extended into the biopsy site, but this was thought to be related to the primary injury rather than the biopsy.

### 3.3. Immunohistochemistry

All biopsy samples collected were of sufficient size and quality to carry out the immunohistochemical analysis. Generally, the IHC procedures in this study were completed, from sample collection to microscopic analysis, within a 36 h time frame. To determine the severity of injury, immunofluorescence staining was used to classify the neuronal injury, dendritic injury, neurovascular injury, and neuroinflammation into 4 grades herein referred to as the Yip, Hasan, and Uff (YHU) grading system. Grade I exhibited limited injury response in comparison to grade IV, which had the most severe injury response. The YHU grading system was devised based on information from existing literature and personal knowledge (PKY).

#### 3.3.1. Neuronal Injury

Neurones from non-injured brain immunostained with NeuN exhibited a large round or polygonal cell body with staining in both the nuclei and cytoplasm, and some proximal processes [15,33,34]. However, neuronal atrophy, vacuoles in the cytoplasm, and neuronal cell loss were also observed after injury [35].

According to the YHU grading system: Grade I, consisting of NeuN+ neurones with limited injury were observed in 8% of patients (Figure 2A, Table 2 and Table 3). Grade II, containing of a few neurones with small vacuoles in the cytoplasm, was observed in 48% of patients (Figure 2B). Grade III, consisting of several NeuN+ neurones with a dysmorphic appearance and/or large vacuolisation within the neuronal cytoplasm, was observed in 20% of patients (Figure 2C). Grade IV, consisting of a limited number of NeuN+ neurones with an atrophic appearance, was observed in 24% of patients (Figure 2D). Overall, the mean (±SD) YHU grade for neuronal injury in this study cohort was 2.60 (±0.96).

#### 3.3.2. Dendritic Injury

Microtubule-associated protein 2 (MAP2) is a specific dendritic marker that has been shown to strongly immunostain dendrites and some nerve cell bodies, and weakly stain axons [29,36]. Moreover, in tissue sections, healthy dendrites exhibit strong MAP immunostaining with smooth and continuous processes. However, after injury, dendrites can develop swelling or a bead-like appearance termed dendritic beading [37].

According to the YHU grading system, Grade I, consisting of MAP2+ dendrites with limited injury response, was observed in 24% of patients (Figure 3A, Table 2 and Table 3). Grade II, consisting of a few MAP2+ dendritic beadings, was observed in 32% of patients (Figure 3B). Grade III, consisting of extensive MAP2+ dendritic beading, was observed in 20% of patients (Figure 3C). Grade IV, consisting of limited MAP2+ dendritic staining with the presence of only a few strands of dendritic beading, was observed in 24% of patients (Figure 3D). Overall, the average YHU grade for dendritic injury in this study cohort was 2.44 (±1.12).

#### 3.3.3. Neurovascular Injury

The tight junction marker claudin-5 [38] and von Willebrand factor (vWF), seen in endothelial cells, [39] shows healthy cerebral microvessels as long regular tubular shapes of approximately less than 10 µm in diameter. In microvascular damage after TBI, fragmented and/or absence of microvessels can be observed [40].

According to the YHU grading system, Grade I, consisting of several regular claudin-5+ and vWF+ long and intact tubular-shaped microvessels, was observed in 4% of patients (Figure 4A, Table 2 and Table 3). Grade II, consisting of many short microvessels, was observed in 32% of patients (Figure 4B). Grade III, consisting of extremely short segments of microvessel, was observed in 36% of patients (Figure 4C). Grade IV, consisting of irregular fragments or absence of any claudin-5+ and/or vWF+ microvessel staining, was observed in 28% of patients (Figure 4D). Overall, the average YHU grade for neurovascular injury in this study cohort was 2.88 (±0.88).

#### 3.3.4. Neuroinflammation

Immunohistochemical markers of microglia with Iba1 and P2Y12 have been widely used in previous studies of neuroinflammation [16,20,41,42]. In healthy brain tissue, microglia exhibit a ramified morphology with long thin processes and a small cell body. However, during neuroinflammation, the processes of microglia become thicker and shorter, and the microglia eventually exhibit an amoeboid shape with retraction of processes [43].

According to the YHU grading system, Grade I, consisting of predominantly Iba1+ and/or P2Y12+ microglia with a ramified morphology, was observed in 16% of patients (Figure 5A, Table 2 and Table 3). Grade II, consisting of many Iba1+ and/or P2Y12+ microglia with shorter and thicker processes, was observed in 36% of patients (Figure 5B). Grade III, consisting of predominantly Iba1+ and/or P2Y12+ amoeboid shape microglia with retracted processes, was observed in 24% of patients (Figure 5C). Grade IV, consisting of fragments and limited (or absence of any) Iba1+ and/or P2Y12+ microglia staining, was observed in 24% of patients (Figure 5D). Overall, the average YHU grade for neuroinflammation in this study cohort was 2.56 (±1.04).

### 3.4. Diagnosis and Prognosis

The main objective for studying fresh brain tissue in TBI was to diagnose the severity of TBI based on the cellular and molecular changes that occurred, and to correlate these with the clinical course and outcome, with the intent of providing an early and accurate prognosis which could then guide clinical decision making. Linear regression was carried out to compare GOS-E with the different YHU injury grades. Linear regression between YHU grade of neuronal injury and GOS-E showed a significant negative correlation (Figure 6A, R^2^ = 0.267, *p* = 0.0082). Furthermore, linear regression between YHU grade of dendritic injury and GOS-E showed a significant negative correlation (Figure 6B, R^2^ = 0.437, *p* = 0.0003). Linear regression between YHU grade of neurovascular injury and GOS-E showed a significant negative correlation (Figure 6C, R^2^ = 0.584, *p* < 0.0001) which was less than the correlation between YHU grade of neuroinflammation and GOS-E (Figure 6D, R^2^ = 0.403, *p* = 0.0007). Interestingly, linear regression between total YHU grade and presentation GCS was not significant (Figure 6E, R^2^ = 0.059, *p* = 0.2416). However, linear regression between total YHU grade and GOS-E did show a significant negative correlation (Figure 6F, R^2^ = 0.610, *p* < 0.0001). Since the injury site varied between patients and the biopsies were collected using two methods, it was important to identify whether this affected the YHU grades. Analysis showed that there were no significant differences, whether the biopsy was proximal or distal to the site of injury (Figure 6G, *p* = 0.7760), or obtained via a craniotomy or before an ICP/EVD insertion (Figure 6H, *p* = 0.1957).

Of the 10 patients with a GOS-E 1 (deceased) or 2 (vegetative state), 100% had a YHU grade III or IV in at least 2 injury types, and 80% had a YHU grade of III or IV in at least 3 injury types (Table 3). However, of the 7 patients with a GOS-E 3 or 4 (severe disability), 57% had a YHU grade III or IV in at least 2 injury types, and 43% had a YHU grade III or IV in at least 3 injury types (Table 3). In contrast, of the 8 patients with a GOS-E >5 (moderate disability to good recovery), 12.5% had a YHU grade III in at least 2 injury types, and 87.5% had only a YHU grade I or II in all injury types (Table 3). In summary, using the YHU grading system, of patients with a total YHU grade of ≥11, 100% of the 11 patients had a GOS-E ≤ 4, which includes death (55%), vegetative state (18%) and severe disability (27%). In contrast, for a total YHU grade of ≤ 8, 100% of the 7 patients had a GOS-E ≥ 5, indicating favourable outcome.

### 3.5. Further Characterisation of the Cellular and Molecular Expressions in the Brain Biopsies

The cellular and molecular expressions in the brain biopsies were further studied for synaptic integrity using synaptophysin, tau phosphorylation using AT8 (detect Ser202 and Thr305 phosphorylation sites), blood vessel permeability using VE Cadherin to detect adherens junction between endothelial cells, and classical anti-and pro-inflammatory microglia using arginase-1 and CD16/32, respectively.

#### 3.5.1. Reduced Synaptophysin Immunopositive Synapses in Severe TBI with Unfavourable Outcome

To study the effect of TBI on synapses, synaptophysin immunostaining was carried out. In normal conditions, synaptophysin immunoreactivity is seen as numerous fine dots in the grey matter [44], and this was observed in all brain biopsy samples. In biopsies from patients with a GOS-E of 7–8 (good recovery), the synaptophysin immunoexpression level (mean ± SEM) was 45,409 ± 8836 A.U. (Figure 7A–C). In comparison to the GOS-E 7–8 group, the synaptophysin immunoexpression level of 27,680 ± 9470 A.U. was non-significantly lower in biopsies from patients with a GOS-E of 4 (moderate recovery) (Figure 7D–F). In contrast, there was a significant reduction in synaptophysin immunoreactivity level in the biopsies from patients with GOS-E 1 (death) (10,254 ± 2126 A.U., *p* = 0.039) in comparison to the GOS-E 7–8 (dichotomised favourable outcome) group, but not compared to the GOS-E 4 (unfavourable outcome) group (*p* = 0.306) (Figure 7H–J). This suggests that there was a significant loss of synapses in severe TBI patients with unfavourable outcomes.

#### 3.5.2. No Change in Tau Phosphorylation in Unfavourable Outcome after Severe TBI

Under normal conditions, tau is located along the axons of neurons, but tau phosphorylation at the Ser202 and Thr205 sites promotes tau mislocalisation to the soma and dendrites [45]. Phosphorylation of tau can be determined using the AT8 antibody and has shown to exhibit distinct phosphorylated tau pathology in post-mortems of patients with chronic traumatic encephalopathy [46]. Interestingly, the lowest AT8 expression (mean ± SEM) was observed in the GOS-E 1 group (3293 ± 748 A.U.), but this was not significantly lower than the mean AT8 expression observed in GOS-E 4 (13,702 ± 6926 A.U., *p* = 0.297), or GOS-E 7–8 (12,446 ± 3311 A.U., *p* = 0.375) groups (Figure 8A–J). This suggests that tau phosphorylation does occur in the brain of severe TBI patients, but it is paradoxically reduced in severe TBI patients with an unfavourable outcome.

#### 3.5.3. No Change in Astrocytic Activation after TBI between Functional Outcomes

After TBI, the intermediate filament glial fibrillary acidic protein (GFAP) has been shown to be elevated in astrocytes and is TBI severity-dependent [15]. Therefore, although the highest GFAP immunoexpression (mean ± SEM) was observed in the GOS-E 1 (44,788 ± 5149 A.U.), it was not significantly higher compared to the GOS-E 4 (25,918 ± 6759, *p* = 0.097) or GOS-E 7–8 (30,261 ± 3215, *p* = 0.203) groups (Figure 9A–J). This suggests that there is no significant upregulation of astrocytic GFAP between the severe TBI groups with different functional outcomes.

#### 3.5.4. Reduction in Endothelial Adherens Junctions in Unfavourable Outcome after Severe TBI

It has been documented that TBI causes cerebral vascular disruption in over 40% of severe TBI patients [47]. As previously mentioned, claudin-5 and vWF, which are markers for tight junction and endothelial cell markers, respectively, have shown negative correlation for severe microvascular damage in patients with a low GOS-E (Figure 4). To further study the microvasculature, the adherens junction between endothelial cells was immunostained with VE Cadherin [48]. In the GOS-E 7–8 group, the VE cadherin expression level (mean ± SEM) was (9380 ± 1090 A.U.), which was higher than the GOS-E 4 group (7524 ± 1236 A.U.). However, there was a significant reduction in VE cadherin immunoexpression for the GOS-E 1 group (5156 ± 310 A.U., *p* = 0.0489). This further suggested that the brain microvasculature is severely damaged in severe TBI patients with unfavourable outcomes.

#### 3.5.5. No Change in Microglia Activation State in Unfavourable Outcome after Severe TBI

To study the activation state of the reactive microglia, the P2Y12 immunopositive microglia were co-immunostained for the classical anti-inflammatory arginase-1 marker and the classical pro-inflammatory CD16/32 marker [20].

##### Classical Anti-Inflammatory Microglia Activation

The arginase-1 immunoexpression levels (mean ± SEM) within P2Y12 immunopositive microglial cells were similar between the biopsies from patients with a GOS-E of 7–8 (7496 ± 3068 A.U.), GOS-E of 4 (5479 ± 587 A.U.), and GOS-E of 1 (7179 ± 4537 A.U.) (Figure 10A–J). In biopsies with a GOS-E of 7–8, there was a low percentage (mean ± SEM) of P2Y12 immunoreactive microglial cells with arginase-1 immunoexpression (15 ± 8%) (Figure 10K). However, there was a non-significant increase in the percentage of P2Y12 microglial cells immunopositive with arginase-1 from biopsies with a GOS-E of 4 (48 ± 11%, *p* = 0.1117) and a GOS-E of 1 (33 ± 10%, *p* = 0.4478) (Figure 10K). Interestingly, there were several strong coexpressed arginase-1 and P2Y12 immunopositive cells with a circular morphology on the periphery of all the biopsies from the GOS-E 1 group (Figure 10L–N).

##### Classical Pro-Inflammatory Microglia Activation

The CD16/32 immunoexpression level (mean +/- SEM) within P2Y12 immunopositive microglial cells was highest in the biopsies from patients with a GOS-E 4 (20,154 ± 7233 A.U.) which was non-significantly higher than the GOS-E 7–8 group (7309 ± 745 A.U.) (Figure 11A–F). However, the GOS-E 1 group (3966 ± 1156 A.U., *p* = 0.0799) was near-significantly lower than the GOS-E 4 group (Figure 11G–J).

Interestingly, the patient group with the highest percentage of CD16/32 and P2Y12 immunopositive microglial cells was observed in the GOS-E 4 group (54 ± 10 %), which was not significantly different to either the GOS-E 7–8 (27 ± 14%, *p* = 0.3280) or the GOS-E 1 (26 ± 13%, *p* = 0.3177) group (Figure 11K). There were a few strong coexpressed CD16/32 and P2Y12 immunopositive cells with a circular morphology on the periphery of all the biopsies from the GOS-E 1 group only (Figure 11L–N).

## 4. Discussion

Currently, there are no effective treatments that have been shown to alter the course of primary brain injury [6]. One potential reason for the failure of clinical trials may be the complex heterogeneity of TBI, especially within the severe TBI group, that is not detected by the simplistic mild, moderate, and severe TBI classification systems based on clinical parameters used to validate these clinical studies [10]. In this study, we showed that fresh brain biopsies from patients suffering TBI classified as severe on clinical grounds could be further subclassified using the YHU grading system into 4 grades of injury severity using immunohistochemistry. In YHU grade I, the neurones, dendrites, microvessels and microglia exhibited a predominantly uninjured appearance. In contrast, YHU grade IV exhibited severe damage to neurones, dendrites, microvessels and microglia, to the extent of absence. Patients with more YHU grade III and IV injuries tended to have unfavourable outcomes, compared to those with more YHU grade I and II injuries, who tended to have favourable outcomes. This information was potentially available within 12 h after injury using a modified immunostaining protocol.

This study further confirmed that a severe closed head injury results in a global response throughout the brain, as has been previously suggested in both animal models [16] and in humans [25]. Performing brain biopsy in living patients is not a new procedure and has been shown to be safe in more than 20 studies over 6 decades without significant complications [23]. Although the safety of brain biopsy in severe TBI has previously been demonstrated [25], we further confirmed that a single-core brain biopsy from 25 living TBI patients did not cause any significant or clinically detectable adverse effects. Single-core biopsies were obtained to mitigate haemorrhage risks, which were fewer than in other studies—in particular, one study with a mean biopsy sample number of 8 in living presenile patients also demonstrated no major adverse effects [22]. Similar to this study, Castejon and colleagues collected cortical brain biopsies during craniotomy from 8 living TBI patients ranging from 1 to 35 days post injury and carried out traditional histological staining [49], successfully identifying brain oedema. Brain biopsy is the cornerstone of neuro-oncological diagnosis, successfully providing a histological diagnosis with low morbidity and mortality [21], and with over 90% accuracy of diagnosis of tumours and lesions in immunodeficient patients [50]. Novel molecular biomarkers have recently added to the diagnostic and prognostic armamentarium in neuro-oncology [51], and we believe that the field of TBI research has the potential to make similar advances.

### 4.1. Injury Markers

NeuN+ neuronal cells have been used in rodents to determine neurodegeneration after TBI [16,33,52]. Therefore, it was not surprising to observe that 92% of severe TBI patients in this study expressed some form of neuronal injury, ranging from small vacuolization (YHU grade II) to major neuronal cell loss (YHU grade IV).

The loss of MAP2+ dendritic staining in YHU grade II–IV dendritic injury was in agreement with the loss of MAP2 staining in human post-mortem brain samples from patients who died of hypoxia-ischemia [36] and within 3 h in a rodent brain with severe TBI model [18]. It has been shown in vitro that mechanical stretch injury to axons (rather than dendrites) causes dendritic beading within 5 min after injury [37]. Interestingly, dendritic beading can also occur due to glucose toxicity [53] and oxygen-glucose deprivation [54]. The critical role of spines on dendrites in synaptic transmission would therefore be lost as a result of dendritic injury, and this would significantly impair neuronal function within the brain [55]. Furthermore, the reduction of the pre-synaptic protein synaptophysin in the patients with GOS-E 1 agreed with observations in moderate TBI in mice [56]. The presence of tau phosphorylation has been detected as early as one day after injury and the level increased with time after TBI in rodents [57]. However, since our biopsies were obtained predominantly within a few hours of injury, it may have been too soon to achieve optimum observation of tau phosphorylation, especially when the biopsy was remote from the site of injury.

The enriched microvasculature in the CNS is tightly bound by the blood–brain barrier (BBB) to provide a physical and metabolic barrier which is essential for normal brain homeostasis [58]. However, after TBI, there is increased vascular leakage due to disruption of the endothelial integrity [59]. The damage and/or loss of claudin-5+ and/or vWF+ stained microvessels in the YHU grade II–IV neurovascular injury suggests that disruption and/or rupture of the neurovasculature occurred in these patients. This was further confirmed with a significant reduction in the adherens junction protein VE cadherin located between endothelial cells in the unfavourable outcome group. Therefore, it suggested that the greater the damage to brain microvessels, the greater the breakdown of the BBB, resulting in oedematous and/or hypoxic-ischemic brain injury [49].

It has been shown that there is a predominant increase in anti-inflammatory microglia/macrophages at 1 day compared to a classical pro-inflammatory response seen at 7 days post TBI in rodents [60,61]. Limiting microglial activation in neuroinflammation has been suggested as a neuroprotective strategy [16,62,63,64]. However, immunostaining for the classical anti- and pro-inflammatory markers ariginase-1 and CD16/32, respectively, in this study, did not show a significant increase in classical microglia activation between the different outcome groups. Therefore, targeting this as a treatment strategy may only benefit a selective patient group with YHU grade II–III neuroinflammation. However, in patients with YHU grade IV neuroinflammation, the detrimental effect after TBI was potentially not microglia activation, but rather the loss of the important normal microglial function [65]. This also applied to the loss of neurones, dendrites, and microvasculature, resulting in an inability to perform their normal roles, which may have contributed to their unfavourable outcome. Interestingly, the astrocytic marker GFAP was not shown to be significantly different between the outcome groups. Although GFAP has been shown to be elevated after TBI [66,67] and has been clinically approved as a serum biomarker for mild TBI [68], our data suggests that GFAP did not contribute to unfavourable outcomes observed in severe TBI.

Of the 10 patients who died or remained in persistent vegetative state, two patients exhibited a maximum of YHU grade III in two injury types, which was not in agreement with the other patients with a similar outcome. The reason for this discrepancy has not been determined, but potential contributory factors included age (the patient who died was within the age range of 76–80 years old) and injury pattern (the patient in vegetative state had a significant injury to the pons).

The loss of any cell type, such as neurones, glial cells, and endothelial cells within microvessels was most likely due to tissue destruction caused by the primary brain injury which seemed to spread beyond the site of impact. In such severe injury, it seems unlikely that any existing therapy would benefit these patients, which accounted for 40% of this cohort. Furthermore, inclusion of these patients could be a potential confounding factor in failure clinical trials in severe TBI [12]. The difference in some patients with neuronal injury, dendritic injury, neurovascular injury, and neuroinflammation suggests that treatments focusing on the predominant injury response are more likely to succeed. Therefore, early diagnosis and prognosis using this cellular and molecular grading system has the potential to provide a significant and accurate additional resource to not only guide patient care but to aid with future clinical trials. One specific clinical situation where brain biopsy may guide clinical decision-making is when determining whether to perform decompressive craniectomy, a highly invasive surgical procedure. Recently, two well-run clinical trials of this lifesaving procedure (DECRA [69] and RESCUE-ICP [70]) came to opposing conclusions which were resolved by consensus which stated the following: “While secondary decompressive craniectomy is a potentially useful operation, it should be applied selectively as there is uncertainty as to which severe TBI subgroups will truly benefit. Decompressive craniectomy may decrease mortality. However, it is not benign and is associated with significant risks of complications and potentially increased risks of disability.” [71].

When decompressive craniectomy is being considered, ICP monitoring would generally be performed prior to decompressive craniectomy providing an opportunity for brain tissue sampling, according to our protocol. The YHU grade could guide decision making on the suitability of decompressive craniectomy, where a low or high YHU grade would support the decision to operate or not to operate, respectively. In this cohort of 25 patients, eight underwent decompressive craniectomy. Four that died had a YHU grade ≥ 11. Of the survivors, two patients had a YHU score of 10: one (aged over 50) had a GOS-E of 2 (vegetative state), one (aged under 30) had a GOS-E of 4. The remaining two patients both had a YHU score of 9 and a GOS-E of 3. Although all these patients would have been dichotomised as poor outcomes in the clinical trials of primary decompressive craniectomy [69,70], none of the patients under the age of 30 with a YHU score <11 died or were left in a vegetative state.

### 4.2. Strengths and Limitations

This study had several strengths. The use of fresh brain tissue obtained from living patients suffering TBI provided a precise and accurate analysis of selective cellular and molecular changes in the injured brain. A very small sample of brain tissue collected as early as 1 h could be used to successfully analyse neuronal and neurovascular injury, and neuroinflammation caused by TBI. This patient cohort consisted of a large variety of head injury mechanisms representative of the variation of head injury types inflicted on the civilian population. The independent histological analyses by PKY and ZHL were blinded to clinical information when determining the YHU grade, and so avoided any bias calls. The immediate fixation of brain tissue resulted in minimal cellular and molecular changes in comparison to post-mortem samples which may take up to 2 days after death before tissue fixation [36], and which necessarily require the patient to die (meaning that it may be days or weeks since the primary injury), and that inadequate perfusion-fixation of a large tissue mass may be responsible for certain false positive pathological observations [49,72].

This study also had several limitations. The 25 patients were all recruited at a single institution (The Royal London Hospital, London, UK). Patients taking anticoagulants, antiplatelet agents, and known alcoholics with likely coagulopathy were excluded to reduce the potential risk of biopsy-induced intracerebral haemorrhage. Brain biopsy from normal healthy control subjects was unavailable, but this would be limited by ethical constraints. It is known that TBI is an extremely complex spectrum of disorders and the YHU grading scale is only an estimate of severity rather than a definitive scoring system. We believe, however, that this could be the inception of a new era of immunohistochemical science in TBI and anticipate that the classification of severe TBI will continue to expand with more TBI-induced specific markers such as neurofilament light protein [73]. While we do not presume to claim that it will eventually provide a definitive answer, we strongly believe that it will add significantly to the armamentarium of tools available to diagnose and prognosticate clinically, and to validate future clinical trials in TBI research.

Future potential applications in low- and middle-income countries are equally exciting as biopsy and immunohistochemical staining may be performed at a low cost and may translate into potential biomarkers for TBI and/or TBI-induced chronic traumatic encephalopathy [73,74]. With standardised staining techniques and web-based upload of micrographs that can be interpreted by artificial intelligence algorithms, this and future grading systems will not just be in the remit of the first world.

## 5. Conclusions

This was the first study to carry out immunohistochemical analysis on fresh brain tissue from living severe TBI patients obtained as early as 1 h after injury. It also showed that it was safe and that multi-pathological factors of varying severity of brain injury could be classified into 4 subgroups using the novel YHU grading system. This YHU grading was shown to be significantly negatively correlated with the GOS-E. Moreover, this study suggested that there could be certain extremely injured TBI patients that would be unlikely to benefit from any current treatments. Using the YHU classification, this information could be made available much earlier than is currently possible through observation of the patient’s clinical course. In summary, we propose that fresh brain biopsies can be used for precise and accurate diagnosis and prognosis of severe TBI. Furthermore, it may be recommended for use in future clinical trials to minimise the number of patients required to demonstrate the efficacy of a particular therapy, removing the confounders of the heterogeneity of the disease and unsurvivable injury, and to maximise the chance of a TBI clinical trial’s success.

## Figures and Tables

**Figure 1 biomedicines-10-00518-f001:**
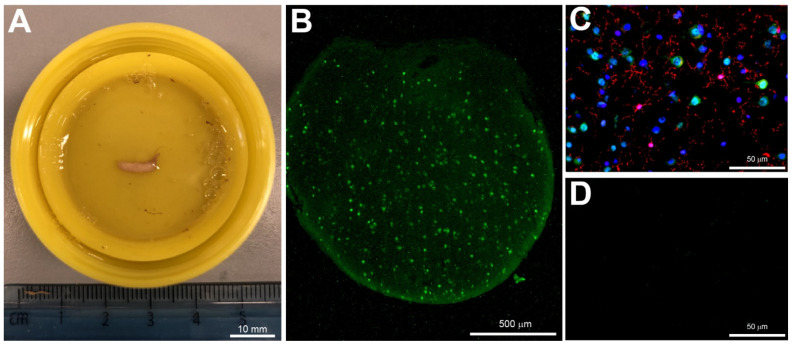
Cerebral biopsy sample and fluorescent immunostaining. (**A**) An image of the cerebral biopsy obtained from a severe TBI patient and fixed in 10% formalin. (**B**) A cross-section of the biopsy stained with the neuronal marker NeuN (green, ×4 magnification). (**C**) Biopsy stained with NeuN (green), Iba1 (red) and Hoechst (blue) (×40 magnification). (**D**) Negative control of immunostaining with the primary antibody omitted. Scale bars: (**A**), 10 mm; (**B**), 500 µm and (**C**,**D**) 50 µm.

**Figure 2 biomedicines-10-00518-f002:**
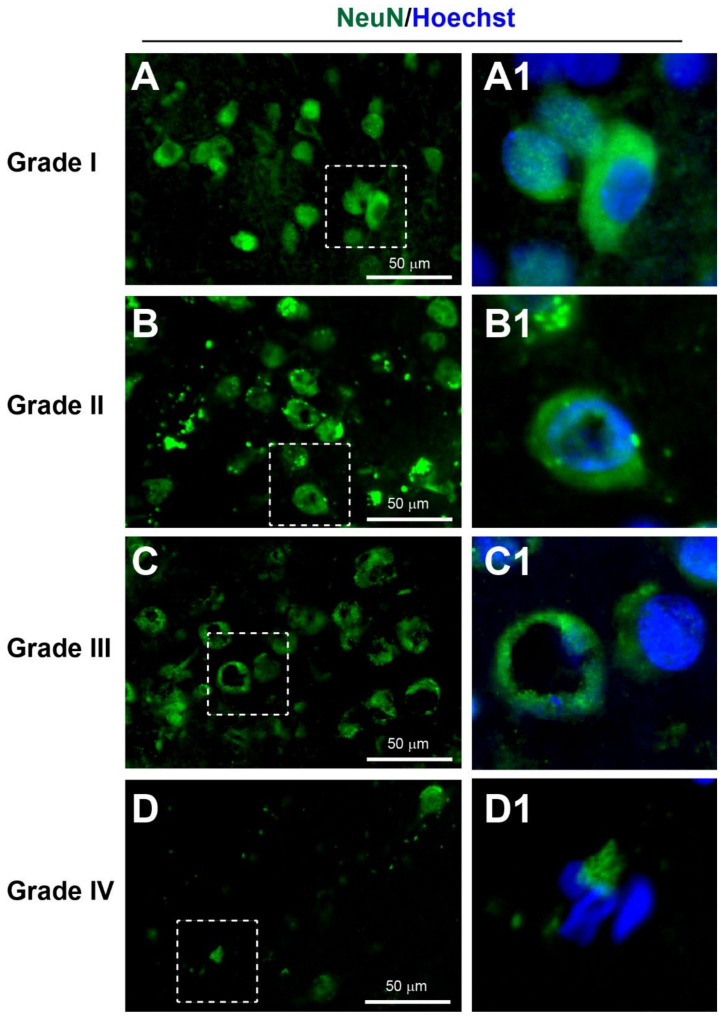
YHU grading system for neuronal injury in cerebral biopsies immunostained with NeuN (green) and Hoechst (nuclei, blue). (**A**,**A1**) Neurones with limited injury display large round/or polygonal cell bodies (grade I). (**B**,**B1**) Neurones with mild damage exhibit mild cellular shrinkage and/or small vacuolisation within the cytoplasm (grade II). (**C**,**C1**) Neurones with moderate damage exhibit moderate cellular shrinkage and/or many with large cytoplasmic vacuolisations (grade III). (**D**,**D1**) Severe neuronal death, indicated by atrophic cell bodies and few, if any, NeuN+ neurones (grade IV). (**A1**–**D1**) Enlarged images of the corresponding dashed boxes. Scale bar is 50 µm.

**Figure 3 biomedicines-10-00518-f003:**
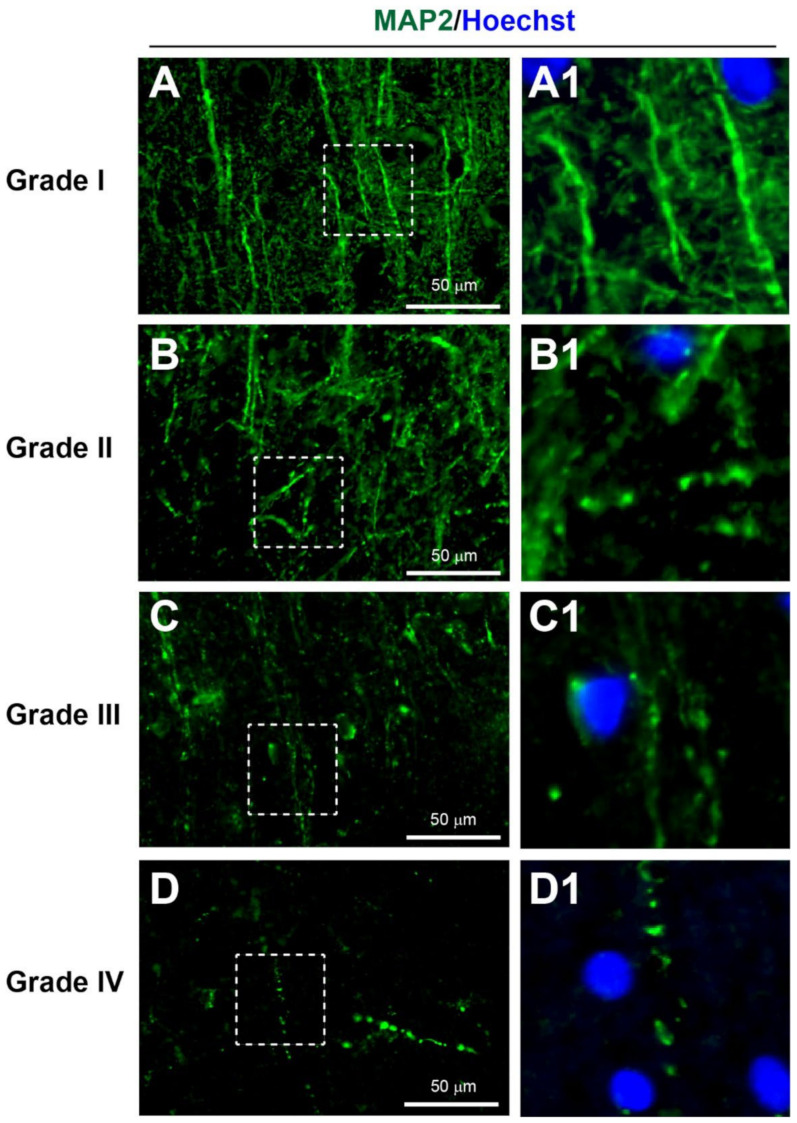
YHU grading system for dendritic injury in cerebral biopsies immunostained with MAP2 (green) and Hoechst (nuclei, blue). (**A**,**A1**) Dendrites with limited injury display strong continuous MAP2+ staining within some cell bodies and along the dendrites (grade I). (**B**,**B1**) Dendrites with mild damage exhibit reduced MAP2 staining and some signs of dendritic beading (grade II). (**C**,**C1**) Moderate dendritic injury exhibited moderate dendritic beading and some complete loss of dendrites (grade III). (**D**,**D1**) Severe dendritic injury is indicated by few, if any, dendrites present and only dendrites with beading present (grade IV). (**A1**–**D1**) Enlarged images of the corresponding dashed boxes. Scale bar is 50 µm.

**Figure 4 biomedicines-10-00518-f004:**
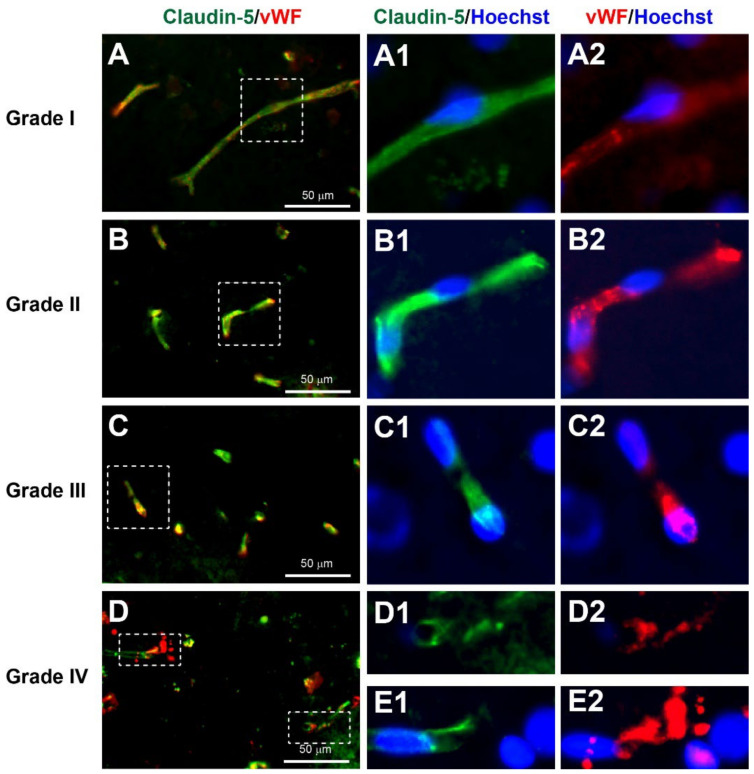
YHU grading system for neurovascular injury in cerebral biopsies immunostained with claudin-5 (green), von Willebrand factor (vWF, red), and Hoechst (nuclei, blue). (**A**–**A2**) Microvessels with limited damage display moderate claudin-5 and vWF staining within long uninjured segments of microvessels (grade I). (**B**–**B2**) Mildly damaged microvessels exhibit stronger claudin-5 and vWF staining, with shorter microvessel segments (grade II). (**C**–**C2**) Moderately damage microvessels predominantly exhibit very short segments (grade III). (**D**–**E2**) Severe microvessel damage is indicated by few, if any, claudin-5- or vWF-stained microvessels present and/or microvessels showing signs of bursting morphology, with release of vWF bodies (grade IV). (**A1**–**E2**) Enlarged images of the corresponding dashed boxes. Scale bar is 50 µm.

**Figure 5 biomedicines-10-00518-f005:**
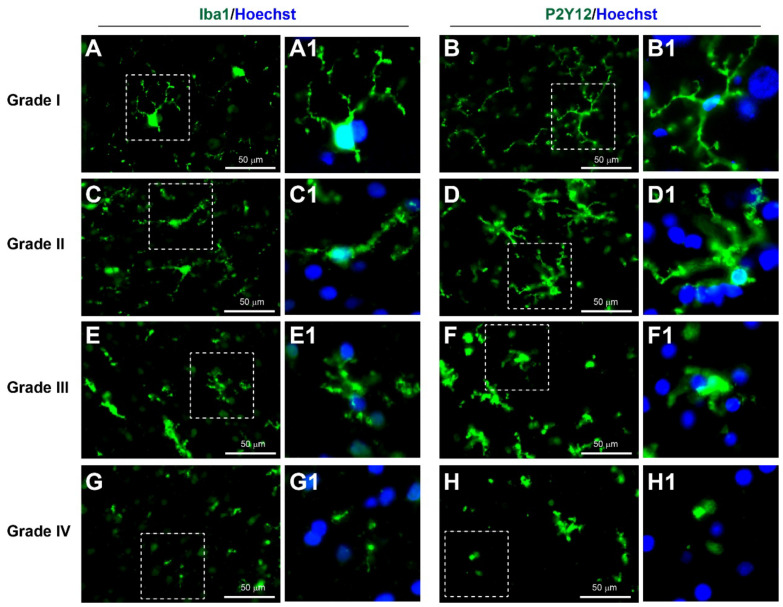
YHU grading system for neuroinflammation in cerebral biopsies immunostained with Iba1 (green, left panels), P2Y12 (green, right panels), and Hoechst (nuclei, blue). (**A**–**B1**) Microglia exposed to limited brain injury display a ramified appearance with long and thin processes and moderate expression of either iba1 or P2Y12 (grade I). (**C**–**D1**) Mild brain injury induces neuroinflammation with increase in microglial Iba1 and P2Y12 expression and thickening of processes (grade II). (**E**–**F1**) In moderate neuroinflammation, the microglia display an ameboid morphology with very short processes (grade III). (**G**–**H1**) In severe brain injury, few, if any, intact microglia are present, and signs of severe fragmentation occur (grade IV). (**A1**–**H1**) Enlarged images of the corresponding dashed boxes. Scale bar is 50 µm.

**Figure 6 biomedicines-10-00518-f006:**
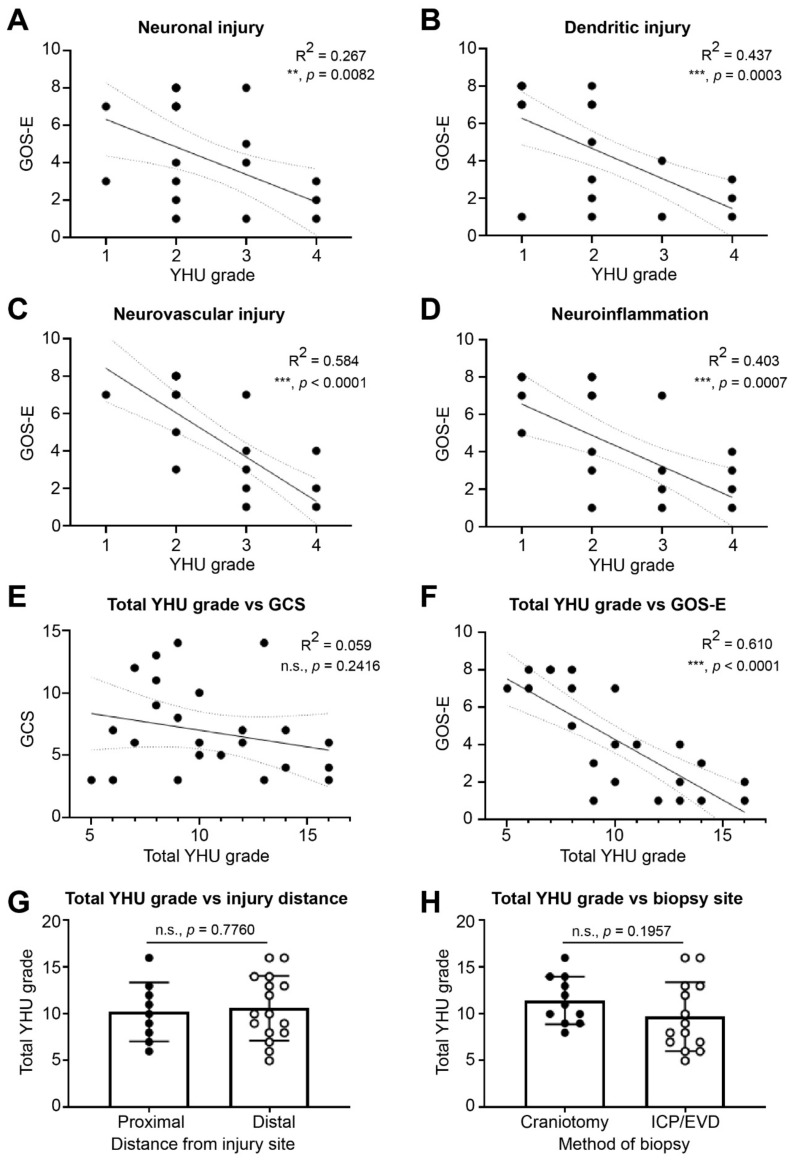
Linear regression analysis between the YHU grade and GOS-E. The GOS-E was compared with the YHU grade for (**A**) neuronal injury, (**B**) dendritic injury, (**C**) neurovascular injury, and (**D**) neuroinflammation. Combination of all YHU grades for neuronal injury, dendritic injury, neurovascular injury, and neuroinflammation of each patient compared with (**E**) GCS, (**F**) GOS-E, (**G**) distance from injury site, and (**H**) method of biopsy. n.s. = non-significant, ** = *p* < 0.01, *** = *p* < 0.001.

**Figure 7 biomedicines-10-00518-f007:**
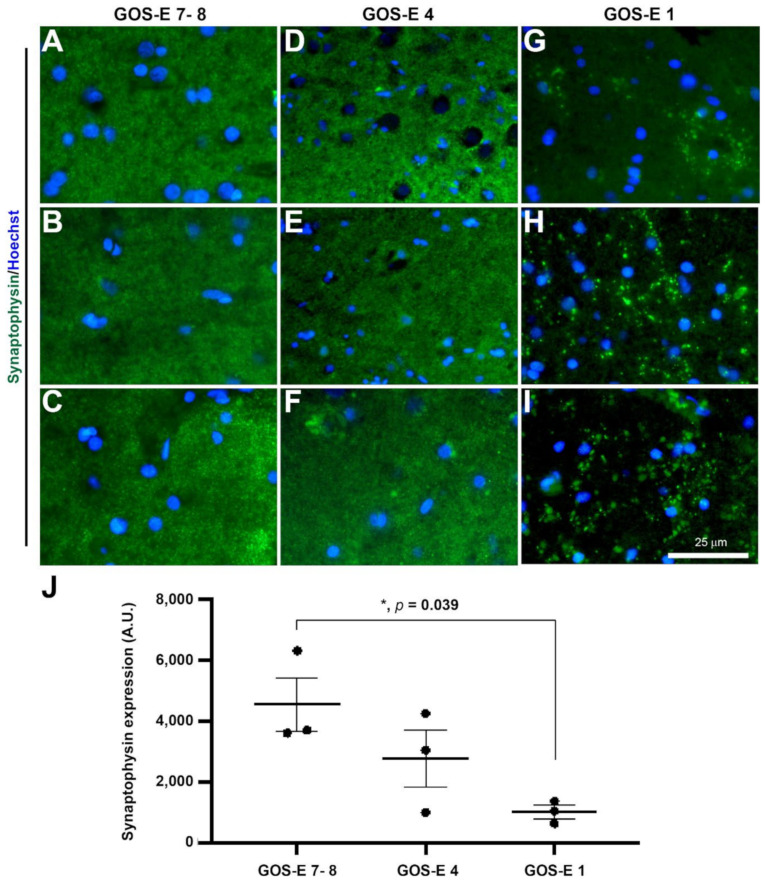
Synaptophysin-immunoreactive synaptic terminal in brain biopsies after severe TBI. Photomicrographs of synaptophysin immunostaining in 3 individual biopsies from patients with a GOS-E 7–8 (**A**–**C**). Photomicrographs of synaptophysin immunostaining in 3 individual biopsies from patients with a GOS-E 4 (**D**–**F**). Photomicrographs of synaptophysin immunostaining in 3 individual biopsies from patients with a GOS-E 1 (**G**–**I**). * *p* < 0.05, GOS-E 7–8 vs GOS-E 1. Results represent mean ± SEM (**J**). N = 3 per group. Scale bar 25 µm. Cellular nuclei stained with Hoechst (blue).

**Figure 8 biomedicines-10-00518-f008:**
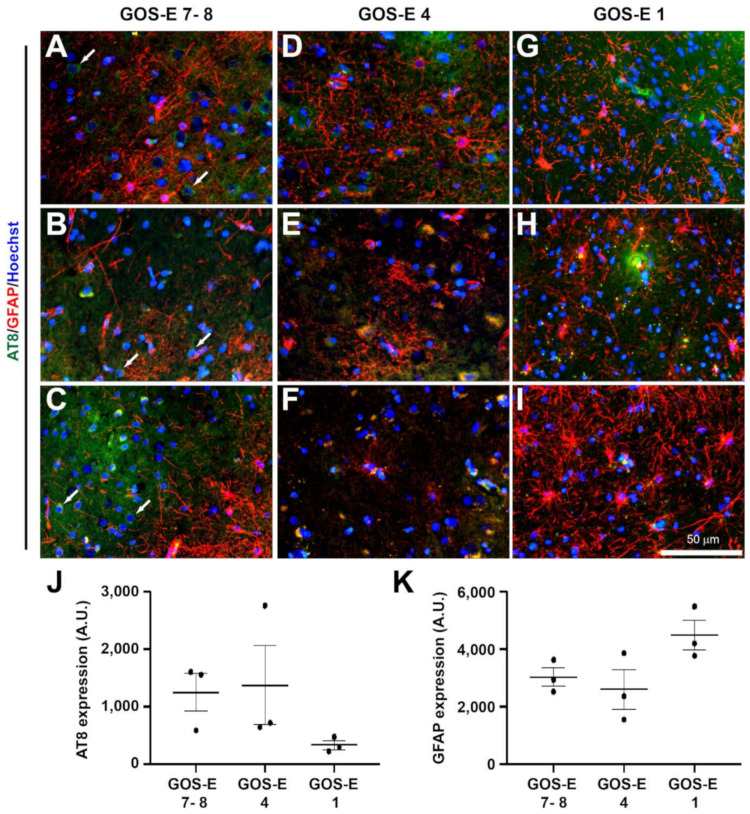
Tau phosphorylation and astrocytic-immunoreactivity in brain biopsies after severe TBI. Tau phosphorylation was stained using AT8 (green) and astrocytes was stained with GFAP (red). Photomicrographs of AT8 and GFAP immunostaining in 3 individual biopsies from patients with a GOS-E 7–8 (**A**–**C**). Photomicrographs of AT8 and GFAP immunostaining in 3 individual biopsies from patients with a GOS-E 4 (**D**–**F**). Photomicrographs of AT8 and GFAP immunostaining in 3 individual biopsies from patients with a GOS-E 1 (**G**–**I**). No significance between immunoexpression in the GOS-E groups was detected for AT8 (**J**) and GFAP (**K**). Results represent mean ± SEM. N = 3 per group. Scale bar 50 µm. Cellular nuclei stained with Hoechst (blue).

**Figure 9 biomedicines-10-00518-f009:**
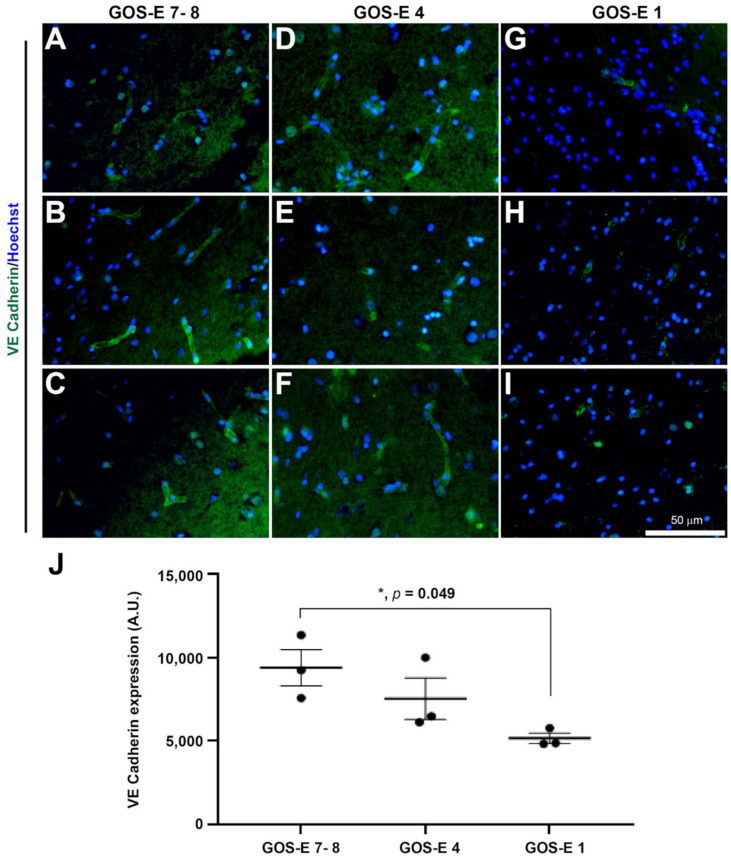
VE cadherin-immunoreactive adherens junctions in brain biopsies after severe TBI. Photomicrographs of VE cadherin immunostaining in 3 individual biopsies from patients with a GOS-E 7–8 (**A**–**C**). Photomicrographs of VE cadherin immunostaining in 3 individual biopsies from patients with a GOS-E 4 (**D**–**F**). Photomicrographs of VE cadherin immunostaining in 3 individual biopsies from patients with a GOS-E 1 (**G**–**I**). * *p* < 0.05, GOS-E 7–8 vs GOS-E 1. Results represent mean ± SEM (**J**). N = 3 per group. Scale bar 50 µm. Cellular nuclei stained with Hoechst (blue).

**Figure 10 biomedicines-10-00518-f010:**
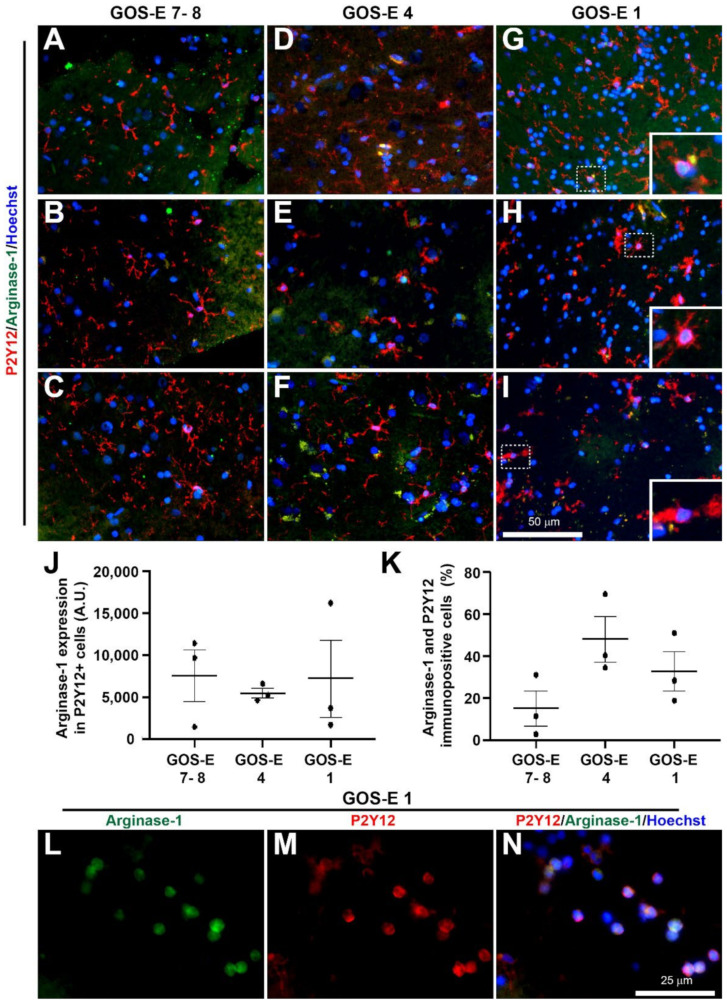
Classical anti-inflammatory arginase-1 marker and P2Y12-immunoreactivity in brain biopsies after severe TBI. Microglia are stained using P2Y12 (red) and the classical anti-inflammatory marker are stained with arginase-1 (green). Photomicrographs of P2Y12 and arginase-1 immunostaining in 3 individual biopsies from patients with a GOS-E 7–8 (**A**–**C**). Photomicrographs of P2Y12 and arginase-1 immunostaining in 3 individual biopsies from patients with a GOS-E 4 (**D**–**F**). Photomicrographs of P2Y12 and arginase-1 immunostaining in 3 individual biopsies from patients with a GOS-E 1 (**G**–**I**). Inserts are enlarged dashed boxes showing cells with P2Y12 and arginase-1 coexpression (**G**–**I**). No significance between immunoexpression in the GOS-E groups detected for overall arginase-1 expression (**J**) and the number of arginase-1 and P2Y12 positive cells (**K**). A few circular cells with both strong arginase-1 and P2Y12 detected, only present in the GOS-E 1 group (**L**–**N**). Results represent mean ± SEM. N = 3 per group. Scale bars 50 µm (**A**–**I**) and 25 µm (**L**–**N**). Cellular nuclei stained with Hoechst (blue).

**Figure 11 biomedicines-10-00518-f011:**
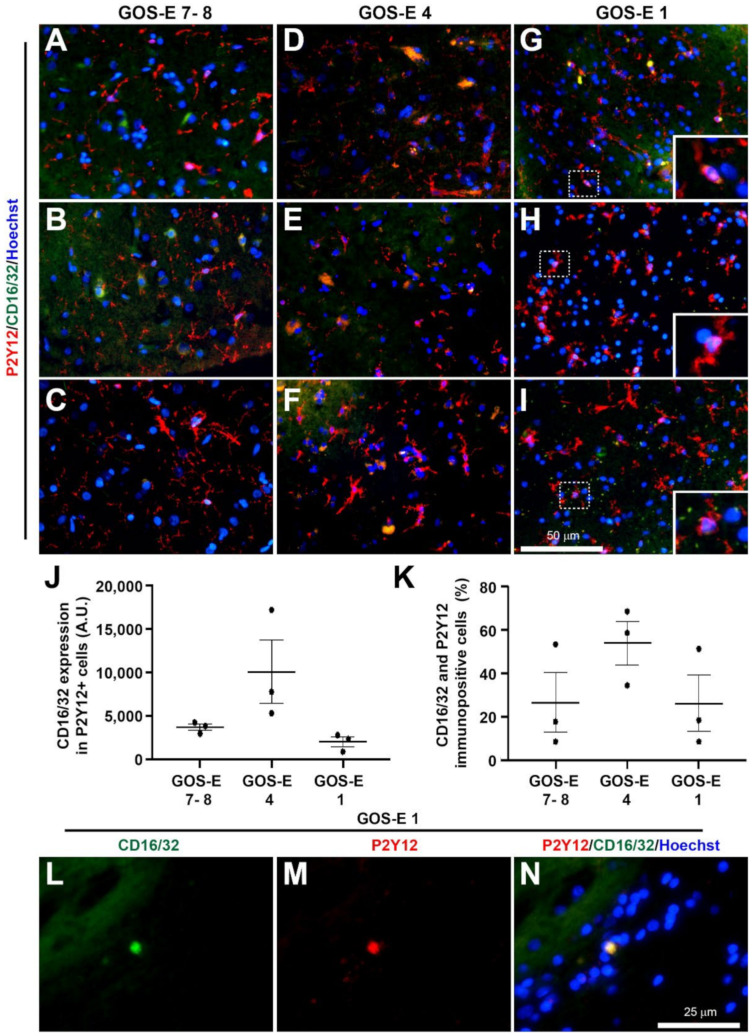
Classical pro-inflammatory CD16/32 marker and P2Y12-immunoreactivity in brain biopsies after severe TBI. Microglia are stained using P2Y12 (red) and the classical pro-inflammatory marker are stained with CD16/32 (green). Photomicrographs of P2Y12 and CD16/32 immunostaining in 3 individual biopsies from patients with a GOS-E 7–8 (**A**–**C**). Photomicrographs of P2Y12 and CD16/32 immunostaining in 3 individual biopsies from patients with a GOS-E 4 (**D**–**F**). Photomicrographs of P2Y12 and CD16/32 immunostaining in 3 individual biopsies from patients with a GOS-E 1 (**G**–**I**). Inserts are enlarged dashed boxes showing cells with P2Y12 and CD16/32 coexpression (**G**–**I**). No significance between immunoexpression in the GOS-E groups detected for overall CD16/32 expression (**J**) and the number of CD16/32 and P2Y12 positive cells (**K**). A limited number of circular cells with both strong CD16/32 and P2Y12 can be detected, but are only present in the GOS-E 1 group (**L**–**N**). Results represent mean ± SEM. N = 3 per group. Scale bars 50 µm (**A**–**I**) and 25 µm (**L**–**N**). Cellular nuclei stained with Hoechst (blue).

**Table 1 biomedicines-10-00518-t001:** Summary clinical details of patients in this study.

SHIBA Code	TBI Type	Age Group (Years)	Gender	Computerized Tomography Imaging	Biopsy Time Post Injury (h)	Trauma Site	Biopsy Site	Biopsy Type	E	V	M	GCS	GOS-E (6 Months)	Injury to Death Duration (Day)
1	RTC (motorcycle)	21–25	Male	Shallow ASDH/ICH/Contusion/tSAH/IVH	146	LF	LF	Craniotomy	1	1	3	5	4	
2	RTC (motorcycle)	21–25	Male	Small EDH/Contusion/DAI	5	RF	RF	ICP	1	1	5	7	7	
3	GSW	16–20	Male	Small ASDH/Contusion/tSAH	1	LT	RF	ICP	2	2	5	9	5	
4	Assault	40–45	Male	Small contusions	7	LF	RF	ICP	3	4	5	12	8	
5	Fall	16–20	Male	Left ASDH	1	L	RF	ICP	1	1	1	3	7	
6	RTC (cyclist)	36–40	Male	Left ASDH/Diffuse contusions/Tight brain	60	LP	RF	ICP	1	1	5	7	1	12
7	Fall	61–65	Male	Left ASDH/LT contusions/All contrecoup	8	RP	LF	Craniotomy	1	1	5	7	3	
8	Fall	46–50	Female	EDH and contusions	1	LP	RF	ICP	1	1	1	3	8	
9	Fall	16–20	Male	DAI	1	RF	RF	ICP	1	1	1	3	1	8
10	Fall	51–55	Male	tSAH and contusions	6	OCC	RF	EVD	3	2	5	10	7	
11	Assault	31–35	Male	tSAH	3	RF	RF	ICP	1	1	4	6	8	
12	RTC (Pedestrian)	56–60	Male	ASDH and DAI	3	RP	LF	Craniotomy	1	2	3	6	2	
13	Fall	46–50	Female	ASDH	48	RF	RF	Craniotomy	4	2	5	11	7	
14	GSW	21–25	Male	GSW	20	LF	LF	Craniotomy	4	4	6	14	3	
15	Fall	76–80	Male	IVH and contusions	1	OCC	RF	EVD	1	2	5	8	1	5
16	Fall	31–35	Male	tSAH and contusions	4	LP	RF	ICP	3	4	6	13	3	
17	Fall	56–60	Male	ASDH	4	RP	LF	Craniotomy	4	4	6	14	4	
18	Fall	26–30	Male	ASDH	2	BLF	RF	Craniotomy	1	1	1	3	3	
19	RTC (Pedestrian)	36–40	Male	ASDH, contusions	1	LP	RF	Craniotomy	1	1	2	4	1	5
20	RTC (Pedestrian)	31–35	Male	Contusions and DAI	3	RF	RF	ICP	1	1	2	4	2	
21	Assault	51–55	Female	Open depressed skull fracture	1	LF	LF	Craniotomy	1	1	4	6	1	33
22	Train accident	56–60	Male	DAI	18	LP	RF	ICP	1	1	1	3	2	
23	RTC (Pedestrian)	51–55	Male	Contusions and DAI	1	OCC	RF	ICP	1	2	3	6	1	2
24	Fall	56–60	Male	ASDH/CSDH	5	RF	RF	Craniotomy	1	1	3	5	4	
25	RTC (Pedestrian)	46–50	Male	ASDH, grade 3 DAI	2	RP	LF	Craniotomy	1	1	1	3	1	4

Abbreviations: ASDH, acute subdural hematoma; BLF, bilateral frontal; CSDH, chronic subdural haematoma; DAI, diffused axonal injury; E, eye; EDH, epidural haematoma; EVD, external ventricular drainage; GCS, Glasgow coma scale; GOS-E, Glasgow Outcome Scale-Extended; GSW, gunshot wound; ICH, intracerebral haemorrhage; ICP, intracranial pressure; IVH, intraventricular haemorrhage; L, left; LF, left frontal; LP, left parietal; LT, left temporal; M, motor; OCC, occipital; RF, right frontal; RP, right parietal; RTC, road traffic collisions; SHIBA, severe head injury brain analysis; TBI, traumatic brain injury; tSAH, traumatic subarachnoid haematoma; V, verbal.

**Table 2 biomedicines-10-00518-t002:** Summary of the morphological characteristics obtained from the immunohistological staining of biopsy samples. The Roman numerals in column 1 represent the YHU grading scales presented in Figure 2, Figure 3, Figure 4 and Figure 5.

YHU Grading Score	Neuronal (NeuN)	Dendritic (MAP2)	Neurovascular (Claudin-5/vWF)	Neuroinflammation (Iba1/P2Y12)
I	Many large round or polygonal cell bodies with uniform cytoplasm ± proximal processes	Many long, smooth and continuous processes surrounded with many thin processes	Several long tubular-shaped microvessels	Small cell bodies with long and thin processes
II	A few large round or polygonal cell bodies with small vacuoles in cytoplasm	Several long processes with some beading surrounded with some thin beaded processes	Several short tubular-shaped microvessels	Small cell bodies with short and thick processes
III	Extensive dysmorphic cell bodies with large vacuoles in cytoplasm	Limited long processes with extensive beading surrounded with limited thin beaded processes	Several extremely short microvessels	Enlarged cell bodies with very short processes
IV	Limited cell bodies with atrophic appearance	Limited beaded processes without any thin processes	Limited short and irregular fragments of microvessels	Limited atrophic cell bodies with cellular fragmentation

**Table 3 biomedicines-10-00518-t003:** Neuropathological examination of fresh brain biopsy samples from the SHIBA patient cohort. The Roman numerals in columns 2–5 represent the YHU grading scales presented in Figure 2, Figure 3, Figure 4 and Figure 5. Total YHU grade is the sum of the grade scores of the 4 injury categories. Patient death or level of disability at 6 months post injury is defined by the GOS-E score.

SHIBA Patient No.	Neuronal Injury	Dendritic Injury	Neurovascular Injury	Neuro Inflammation	Total YHU Grade	GOS-E
1	II	III	III	II	10	4
2	I	I	II	II	6	7
3	III	II	II	I	8	5
4	III	I	II	I	7	8
5	II	I	I	I	5	7
6	IV	III	III	II	12	1
7	IV	IV	III	III	14	3
8	II	I	II	I	6	8
9	III	IV	IV	II	13	1
10	II	II	III	III	10	7
11	II	I	II	II	7	8
12	II	II	III	III	10	2
13	II	II	II	II	8	7
14	I	II	II	IV	9	3
15	II	I	III	III	9	1
16	II	II	II	II	8	8
17	II	III	IV	IV	13	4
18	II	II	III	II	9	3
19	IV	III	IV	III	14	1
20	IV	IV	IV	IV	16	2
21	III	II	III	IV	12	1
22	II	IV	IV	III	13	2
23	IV	IV	IV	IV	16	1
24	III	III	III	II	11	4
25	IV	IV	IV	IV	16	1

## Data Availability

Not applicable.

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
