# Peer review of "Characterisation of Severe Traumatic Brain Injury Severity from Fresh Cerebral Biopsy of Living Patients: An Immunohistochemical Study"

_biomedicines, 2022, doi:10.3390/biomedicines10030518_

Round 1

Reviewer 1 Report

This is a well designed and conducted study.  Introduction provides relevant information. Methodology is appropraite. Results are well displayed and discussion is balanced. Refernces are adequate.

Author Response

Thank you for the positive comments. There was no further points to address.

Reviewer 2 Report

The current study investigated severity of TBI by applying IHC on fresh brain biopsy and they concluded that based on the cellular and molecular pathophysiology according to the YHU grading system, they could classify the severity of injury. It is an interesting study but there are some issues that need to be considered:

  1. In the abstract, it is not clear what is the advantage of the method that authors used, what they found by applying the new method that they could not find it before.
  2. In introduction, it is mentioned that classification of TBI is not match with the outcome due to the incorrect classification, however, the outcome of TBI depends on several factors like age and comorbidity, how authors would discuss about that?
  3. What is the advantage of microscopic investigation of tissue to the imaging methods such as MRI and CT for classification of TBI severity?
  4. How authors reached to the point to get biopsy from the superior frontal gyrus and not from other parts of brain? Does only one biopsy from one brain region gives reliable information for classifying the TBI severity comparing with imaging methods which give the overview of whole brain?
  5. In figure 1C , red staining does not look like microglia rather looks stained parts of vessels.
  6. There is no information about the comorbidity which would have significant influence on the outcome.
  7. In the result section, authors did linear regression analysis to correlate brain injury with the outcomes, but this is not enough to find the suitable biomarker for early diagnosis. It is important to group the patients base on their grades and do comparison between them based on the outcome. The other thing is that outcome is based on the GOS-E score so it needs to be grouped based on the scores and compare the grades between them.
  8. It is not clear which statistical test was used for each analysis.
  9. Was there any correlation between the outcome, injury reason and age?

Author Response

The current study investigated severity of TBI by applying IHC on fresh brain biopsy and they concluded that based on the cellular and molecular pathophysiology according to the YHU grading system, they could classify the severity of injury. It is an interesting study but there are some issues that need to be considered:

  1. In the abstract, it is not clear what is the advantage of the method that authors used, what they found by applying the new method that they could not find it before.

Thank you for taking time out to review this maunscript. The abstract has been updated (lines 14-27).

  1. In introduction, it is mentioned that classification of TBI is not match with the outcome due to the incorrect classification, however, the outcome of TBI depends on several factors like age and comorbidity, how authors would discuss about that?

This has been addressed in the introduction (lines 51-53) and with the addition of two references (Ponsford (2013) ref 3 and Cole et al., (2015) ref 7).

  1. What is the advantage of microscopic investigation of tissue to the imaging methods such as MRI and CT for classification of TBI severity?

The advantage is that medical imaging lacks resolution to detect cellular and molecular changes. The authors respectfully suggest that this is addressed specifically in lines 71-79. No changes have been made.

  1. How authors reached to the point to get biopsy from the superior frontal gyrus and not from other parts of brain? Does only one biopsy from one brain region gives reliable information for classifying the TBI severity comparing with imaging methods which give the overview of whole brain?

This is now addressed in lines 90-97: A single study performing brain biopsy in the superior frontal gyrus (the standard position for ICP monitors) prior to ICP monitor insertion in severe TBI demonstrated the safety of the procedure [22], and in addition to this study which demonstrated a global TBI associated proteomic response, there is evidence in animal models that severe TBI results in global biomarker (Galectin-3) expression throughout the brain [14]. We therefore aimed to investigate the global cellular and molecular response to TBI within the brain using a standardized location; biopsies obtained at craniotomy were also taken from the superior frontal gyrus.

  1. In figure 1C , red staining does not look like microglia rather looks stained parts of vessels.

This figure 1C has been changed to avoid any ambiguity.

  1. There is no information about the comorbidity which would have significant influence on the outcome.

This is now addressed in lines 207-212: The patients had very few comorbidities and the protocol appears to have introduced an unintentional bias towards this by excluding those taking anticoagulant or antiplatelet medication. All the patients who did not survive died from their head in-jury. A single patient also had COVID pneumonia (as will be discussed below) but apart from this there were no patients where comorbidity as opposed to TBI was considered to have contributed to the outcome.

  1. In the result section, authors did linear regression analysis to correlate brain injury with the outcomes, but this is not enough to find the suitable biomarker for early diagnosis. It is important to group the patients base on their grades and do comparison between them based on the outcome. The other thing is that outcome is based on the GOS-E score so it needs to be grouped based on the scores and compare the grades between them.

The linear regression compares two independent variables (YHU vs GOS-E) to validate them in this patient population. Our aim is to demonstrate the individual pathologies associated with different injury types and therefore placing patients into groups is counter-productive. Furthermore, it is important to note that the unfavourable outcome may just involve one pathology (e.g. damage of microvasculature) rather than a cummulation of multiple pathologies, even though latter is common.

  1. It is not clear which statistical test was used for each analysis.

This is detailed in lines 185-187.

  1. Was there any correlation between the outcome, injury reason and age?

There was no correlation between outcome and injury reason. Age is known to be a contributory factor which is not considered in current TBI grading systems and this has been addressed in the response to point 2 raised by this reviewer. Age has deliberately not been considered as this study aims to investigate the cellular and molecular changes.

Reviewer 3 Report

Interesting paper looking at brain biopsy with IHC assessment to grade severity of injury and correlate with outcomes. Could be of good clinical utility, especially in poor resource countries. 

Figure 1 is sufficient and shows the standard biopsy core. 

Table 1 is helpful to look at overall demographics. 

In table 2, the YHU grading is interesting. Would additional markers such as iNOS, Arg1, GFAP, Occludin, and tau markers be helpful? The grading system needs to be expanded as outlined below. 

Table 3 is sufficient. 

Figure 2. It would be helpful to see synaptophysin staining. 

Figure 3. Please stain for marker of tauopathy as well. 

Figure 4. Please stain for occludin as well. 

Figure 5. Would be helpful to characterize microglia by staining iNOS, Arg1, and Cox-2. 

Figure 6 data is convincing and could be strengthened with above mentioned assays. 

Some key references are missing and should be included: PMID: 23335911; PMID: 34901937; PMID: 26140712

Author Response

Interesting paper looking at brain biopsy with IHC assessment to grade severity of injury and correlate with outcomes. Could be of good clinical utility, especially in poor resource countries. 

Thank you for taking time out to review this manuscript.

Figure 1 is sufficient and shows the standard biopsy core. 

Thank you.

Table 1 is helpful to look at overall demographics. 

Thank you.

In table 2, the YHU grading is interesting. Would additional markers such as iNOS, Arg1, GFAP, Occludin, and tau markers be helpful? The grading system needs to be expanded as outlined below. 

Thank you. Although the grading system is potential not complete as mentioned in the discussion (lines 871 - 874), the authors do not want to complicate the first ever YHU grading system. However, we appreciate additional markers would strengthen the manuscript, so we have carried out further immunohistochemical staining on the brain biopsies as requested. To detract from the initial findings and the YHU grading system, we have organised the additional staining as a ‘further characterisation of the brain biopsies’.

Table 3 is sufficient. 

Thank you.

Figure 2. It would be helpful to see synaptophysin staining. 

We have provided a new figure 10 showing the synaptophysin staining.

Figure 3. Please stain for marker of tauopathy as well. 

We have provided a new figure 11 showing the AT8 staining.

Figure 4. Please stain for occludin as well. 

We have provided a new figure 12 showing the VE cadherin staining. Given that claudin-5 and occludin are tight junction markers, the authors preferred to stain for another endothelial cell barrier marker, which is VE cadherin, an adherens junction marker to demonstrate endothelial junction integrity.

Figure 5. Would be helpful to characterize microglia by staining iNOS, Arg1, and Cox-2. 

We have provided two new figure 12 and 13 showing the arginase-1 (classical anti-inflammatory marker) and Cd16/32 (classical pro-inflammatory marker) costained with P2Y12 microglia marker.

Figure 6 data is convincing and could be strengthened with above mentioned assays. 

We have provided the additional immunostaining as requested above by this reviewer.

Some key references are missing and should be included: PMID: 23335911; PMID: 34901937; PMID: 26140712

 We have included the missing references as requested. These references are numbered 74, 73, 47 on the reference list.

Round 2

Reviewer 2 Report

The authors replied to the most of the comments in a satisfactory manner, one comment still needs to be replied.

It is not clear which statistical test was used for each analysis.

This is detailed in lines 185-187.

The information that authors added are related to the details of quantification of cells , while the comment is  about the applied statistical method,. and also this information about cell counting should be moved to the method section 

Author Response

Apologies for misunderstanding the comment. We have now provided further information related to the statistical analysis and have moved the ImageJ cell analysis paragraph to the 2.3 immunohistochemistry section as requested.

Reviewer 3 Report

Addressed all issues and ready for publication. Recommend acceptance. 

Author Response

Thank you for the acceptance.